environmental science

organic agriculture, new ruralities, rural networks

**Author for correspondence:**
Nádia Jarouche Aun
e-mail: nadiarpe@gmail.com

Contribution to Sustainable Land Use collection.

# Organic agriculture and rural networks in the mountain environments of Região serrana fluminense, Rio de Janeiro, Brazil

Nádia Jarouche Aun[1] and Renato Linhares de Assis[2]

[1]Program in Science, Technology and Innovation in Agriculture from Universidade Federal Rural do Rio de Janeiro (UFRRJ), Seropédica, Rio de Janeiro, Brazil
[2]Embrapa Agrobiology, Professor and Tutor at UFRRJ PhD program Science, Tecnology an Agricultural Innovation, Seropédica, Rio de Janeiro, Brazil

NJA, 0000-0003-3410-5719

This paper presents a case study on organic agriculture at Região Serrana Fluminense, Rio de Janeiro State, Brazil. We sought to understand what was the role of organic farming, and if it can be considered as an aggregating element between different kinds of people or groups who were located in the same region. The methodology used to carry out this investigation was based on the concept of the rural network, which enabled us to comprehend the articulations and connections that comprise the organic agriculture circuit in the mountains of this specific region.

## 1. Introduction

The purpose of this research was to highlight aspects of local groups involved with organic farming in Região Serrana Fluminense. We wanted to understand how the consolidation of small groups involved in organic farming could contribute to the strength of organic production. We attempt to search for the elements that contributed to the consolidation of the groups and the practice itself, such as tourism or a new form of validation to the organic production.

This particular region comprises 16 municipalities, all of them present similar climate and geographical characteristics. The local main economic activity is agriculture, except for Nova Friburgo, Petrópolis and Teresópolis, which also stand out in the services and industry sectors. The development of organic farming in this region came about during the 1980s, and it was consolidated at the end of the following decade, thanks to an

alliance between agronomists and traditional local farmers who saw in the local mountain environment an ideal place to build a new way of life associated with a new way of doing agriculture.

In 2018, there were 186 organic farmers (considering only Petrópolis, Nova Friburgo and Teresópolis) organized in associations, groups or companies in the region with the purpose of producing and marketing food. Even though these associations are outnumbered by other rural producers of the region, they can also count on the knowledge of different types of professionals: from lawyers to chefs, those organic groups have developed a special way to establish a connection across the various regions where organic food is present.

We have focused the present article on organic rural workers from Nova Friburgo and Teresópolis. First, we contextualize the research. Then, we present the methodological procedures used to identify the rural network. Finally, we depict actors (we prefer to use the term 'actors' because it refers to all people involved in the community/group/town including 'stakeholders'), dimensions and how they are organized around organic agriculture and how they are contributing to institute a more complex rural network.

## 2. Teresópolis and Nova Friburgo

Rural production and tourism are common economic activities between Nova Friburgo and Teresópolis. Rural production has always been a local aptitude, although the territory faces restrictions to mechanization and difficulties installing a production flow. The mild climate (tropical of altitude) with rich vegetation cover (Atlantic Forest) and year-round high rainfall has allowed the region to establish itself as the green belt of Rio de Janeiro, the capital and the metropolitan region.

The properties are small in size and the predominant crop is of vegetables [1]. This region's agricultural production accounts for 90% of the vegetable supply [2] for the 12 377 505 inhabitants [3] of the metropolitan region of the city of Rio de Janeiro.

Pluriactivity was identified as a unique feature regarding the actors from the region. Marafon & Ribeiro [4] and Carneiro & Rocha [2] have observed the existence of farmers who had other sources of income both in urban centres and within their properties, which were chiefly related to tourism. However, there is the perception that the importance of agricultural activities in the composition of household income is decreasing, nonetheless it will not disappear. Agricultural activities are part of the identity of the population and therefore the territory [2].

According to the aforementioned authors, farmers' behaviour is a very important attribute for the economic performance of families as well as of the localities where they live. In Teresópolis and Nova Friburgo, besides agricultural activities, tourism and industry are important economic sectors. Therefore, farmers pluriactivity tend to generate more income and satisfaction towards rural activity once they are not only dependent on one kind of income, strengthening families and thus the region where they are established.

Just as the mountains attracted a type of tourism as an alternative to the coastal regions of the state, it also contributed to the creation of another identity to the region. Between the municipalities of Nova Friburgo and Teresópolis, there is a great touristic movement enhanced by gastronomy and sustainability, which has found in this space a great ally to spread its philosophy and build a new standard for food consumption, promoting organic farming.

## 3. Organic agriculture at Região Serrana Fluminense

Agricultural production is a striking feature of the landscape at Região Serrana Fluminense. It is responsible for supplying most vegetables consumed in the state capital. However, their production and trade processes still resemble an industry, that is to say, large-scale output, reduced prices and a strong dependence on external inputs to the agricultural production unit. It is a model adopted in the late 1960s that continues to exist in the present day in the region, representing the majority of food yield in those areas [1,5].

The beginning of organic agriculture in the region was at the end of the 1970s [6]. In fact, during those years, in some regions of Brazil, there was a movement opposing the conventional model of food production strongly linked to an industry. The group of people involved in this movement proposed a new form to cultivate food, a more sustainable alternative to the conventional model. However, the institutionalization of the term organic agriculture only went further in 2003 with specific legislation, although its regulation had already been discussed since the beginning of the 1990s [6].

According to Felippe [6], the diffusion of organic production in Região Serrana Fluminense happened with the support of professionals related to agrarian sciences and was one of the first in Brazil. Its diffusion in the region is owing to the work of some professionals in the field, but also to the articulation and creation of the Association of Biological Farmers of Rio de Janeiro (ABIO). This association, besides fomenting the practice of organic agriculture and disseminating specific knowledge regarding organic food production, was also an important element throughout the process of the construction of the Brazilian legislation on organic production. One of the most important achievements is the Brazilian System of Organic Conformity Assessment (SISORG), a pioneer until now in the way in which the conformity of organic production in the country is evaluated.

SISORG was elaborated in 2010, and since then it has recognized three ways to guarantee the conformity of organic production in Brazil. The first one is by auditing, the most common in the world, a company plays the role of auditing the productive areas. The second format is through social control among the farmers themselves. It is a model used only for direct sale to consumers; a model more restricted to the size of the production unit, and the way in which food is sold. The third way of ensuring production conformity is through a participatory guarantee system (PGS). A little more complex than the social control, this system has the same equivalence of the conformity guarantee made through auditing.

Offering other ways to ensure the conformity of organic production, besides auditing, gave rise to an increase of institutionalized productive units. According to Felippe [6], in April 2010, ABIO had 11 PGS groups and by June 2017 there were 39 groups. This innovative and pioneering format is also characterized by being much more accessible and inclusive for producers who wish to start the process of obtaining an organic label for their production and still have some insecurities regarding their processes.

Other members of the community, such as consumers, technicians, must form a PGS group. They become active members, with the obligation to attend meetings and verifications in the productive units. It is within this socially constructed space where the exchanges (flows) of diverse knowledge and the building of trust among the members of the group are sought.

# 4. Organic agriculture: who practices and who consumes

In order to understand who are the actors involved with organic agriculture in this particular region, it is important to start from the concept of neorural population, which is considered as a social category destined to the definition of an individual within a society. From that concept, it was possible to arrive at the new rurality points of view, which represent the paradigm shift within spaces where the neorurals or new actors are gathering.

With a wider perspective, the new ruralities represent a new way of perceiving the localities, considering social relations, environment exploration and the political and economic culture. It is a broader concept that considers all the actors involved and the spaces where they are acting and transforming. For Brandenburg [7], the concept of new ruralities, more specifically, the reconstruction of a new rurality is the result of interaction through the search for new life options, usually related to activities involving the natural environment. This involvement among groups of different origins in the rural environment became possible by the multifunctional character that the field presents, enabling the improvement of activities geared to agricultural production, industry, tourism and rural recreation [7].

Mattos [8] relates the appearance of organic agriculture in the southeastern state of Rio de Janeiro to the arrival of neorural populations in the region. According to the author, the reverse migration movement to the countryside took place mainly because of the dissatisfaction experienced by these actors in large urban centres and the search for more quality of life. The author also reports it was a movement that, in the early 1990s, was more closely related to the quality of life, and during the 2000s, it also became an economically viable opportunity.

At the same time, in the early 2000s, there was a greater appreciation of more sustainably produced foodstuffs in large urban centres and they became more representative in local street fairs and distribution centres, resulting in an increase in the demand for organic foods. The popularization of organic agriculture (among consumers) and the perception of greater profitability (among farmers) are factors that also influenced the conversion of local and conventional farmers to an organic system of cultivation.

Thus, this network that started to inhabit the rural area in the region, had as its first characteristic the ideology of alternative production and environmental protection. Over the years, other factors have been incorporated into this network, and what was just an ideology became an expanded range of possibilities and now encompasses a great heterogeneity of types.

# 5. Tourism in Região Serrana Fluminense and its relationship with organic agriculture

Tourism, like agriculture, has a wide range of possibilities and definitions. When we consider non-industrial agricultural production, for example, we have biodynamic agriculture, natural agriculture, organic agriculture, and permaculture, among others. The same happens with tourism, especially when trying to define rural tourism. There is tourism in rural areas; rural tourism; agrotourism; ecotourism and ecological tourism; and adventure tourism, among other modalities.

According to Soares [9], agrotourism emerged in Europe in order to compensate for a decaying agricultural activity, as well as to combat rural exodus and cultural erosion. Both themes are frequently found in research about the countryside of Portugal, portrayed by Fonseca [10]. The focus on tourism under these conditions relies on providing the residents with more options to integrate with the local economy and more opportunities for development rather than assisting the exodus of their inhabitants to seek opportunities elsewhere.

In Brazil, agrotourism is still little explored. The most common modalities found by Soares [9], related to agriculture itself, are those in which the tourist only enjoys the property for one day, either with the daily practice of the productive unit or the experience of a day in the countryside, with property tours and typical food. Thus, more generally, rural tourism or tourism in rural areas is one in which we can also count on some expressiveness of agricultural activities, even if it is in the form of visitation to local free markets in the urban centres of municipalities.

According to Marafon & Ribeiro [4], tourism at Região Serrana Fluminense, also considered as 'Top of the mountains tourism' by the authors, includes mainly the municipalities of Petrópolis, Teresópolis and Nova Friburgo. It emerged as an alternative to tourism practised in the coastal region of the state. The authors classify this region as a place for the development of contemporary rural tourism, largely populated by hotels, spas and restaurants, in a context that highlights the importance of organic agriculture.

Regarding this last topic, in organic agriculture, according to the Ministry of Agriculture, Livestock and Food Suply's [11] National Register of Organic Production, it was more frequent to encounter production units (with organic seal) closer to places of greater tourist appeal. Some explanations are based on the proximity to places of greater environmental preservation, landscape appeal and the environment that provides leisure, relaxation or even an option for exploration of wilder nature. To reinforce that theses, we found in the data provided by the Ministry that 31% of 723 organic units were based in the Região Serrana Fluminense, that number represents 186 organic production units based between the municipalities of Nova Friburgo, Teresópolis and Petropolis.

Just as the mountains attracted a type of tourism that sought an alternative to that practised in the coastal regions of the state, it also contributed to the creation of another identity for the region. It is between the municipalities of Nova Friburgo and Teresópolis that concentrates great tourist movement combined with gastronomy and sustainability. Thus, organic agriculture found in this space is a great ally to spread its philosophy, sell its products, and build a new pattern of food consumption.

Within a new approach to rural development, which considers endogeneity as one of its dimensions, it is possible to point out the relationship between tourism and organic agriculture as a benefit to the locality. This approach tends to contribute positively to the shaping of local rural networks by expanding the possibilities of relationships between actors, with an increase in the number of people within groups, just as it favours the construction of autonomy for the region linked to its own identity.

# 6. Research methods and procedures

To compose a comprehensive analysis of the region where the case study was conducted, as well as the groups surveyed, we gathered methodological elements adopted by the qualitative research, the multilevel approach and the rural network methodology. In addition to these procedures, we also analysed local documents and carried out a bibliographic review regarding the content.

The methodology of the rural network is based on the re-signification of development. The fundamental concept of the development comprises six theoretical dimensions: social capital, endogeneity, sustainability, market governance, innovation and institutional arrangements. This approach was developed by authors Jan Dowe van der Ploeg and Terry Marsden in *Unfolding webs: the dynamics of regional rural development* [12]. The purpose of the book is to reflect and produce concepts aimed at the construction of a new theory about development based on the understanding of rural networks.

Rural network methodology's proposal is precisely to study territories from a standpoint that considers the most varied levels of a social organization. The multilevel approach adds to this the idea of understanding with more detail, the relations that are being built between the actors within the web. With both methodologies is possible to understand with a more clear perspective of how actors relate inside the territory, or even create their own territory.

All the instruments used during the survey had the objective of capturing the variety of dimensions within each group studied and not only the economic capacity of organic agriculture to sustain a certain community. The dimensions suggested by Ploeg & Marsden [12] are, in fact, concepts defined during a long research process and were defined as the primary structure of a rural network.

We started with the definition of how contacts with farmers, technicians, administrators, local entrepreneurs and other local 'types' would be made. Because the analysis would be based on the existence (or not) of social networks, we opted for the snowball approach to compose the group of interviewees.

Snowball sampling has a main characteristic of the collection of data in a non-probabilistic way. This approach relies upon the researcher's ability to obtain one (or more) key informant (or seed) and from that person build a network of contacts. According to Vinuto [13], it is a method that uses reference chains and is indicated for taking samples in networks, especially when it comes to groups that are difficult to access because their components are far away from each other, have a more reclusive behaviour, or the informants are unwilling to take part in the research.

As defined by the methodology, we used key informants to approach the groups, thus forming our universe of research made up of 24 rural producers throughout Teresópolis, Nova Friburgo and neighbouring municipalities; two EMATER-RIO technicians; three ABIO technicians and four local entrepreneurs who work with gastronomic tourism.

We chose to construct a semi-structured questionnaire for the interviews. The purpose of our questions was to investigate the relationships that were being built and strengthened (or not) within the groups and how such relationships influenced (or not) the constructions of a more complex network of relationships around organic farming. In this way, we tried to create a road map that could cover the six dimensions of the rural network, seeking information about: general characteristics of the farmers and their production; endogeneity; sustainability; social capital; institutional arrangements; innovation and entrepreneurship and governance.

The most important indicators we defined to interpret the questions made in the semi-structured questionnaire were: how often the same reference (local farmer) was given to us during an interview; the frequency observed in the meetings of validation of the organic production; and how all members of the group communicate between each other (how the power structures were more or less expose between the actors). Other kinds of relationship that we find important to understand was the way they try to participate in the governmental process or how they define a better way to commercialize their products. At last, we found it important to understand the different origins of each interviewee and its former activity, the intention was to understand the possibilities of bringing some new aspects (knowledge/capacity) to the web.

The questionnaire gave us a great volume of data but participant observation played a crucial part during the research. Participant observation, according to Peruzzo [14], is a widely used instrument within the universe of qualitative research, its main function is to observe the groups studied without, however, being an invisible part of the process. In the context of this research, this tool allowed participation in street fairs, association meetings and technical field trips.

It is important to point out that the dimensions suggested by Ploeg & Marsden [12] in the rural web methodology were perceived during the development of long research that involved a great number of academic researchers in various countries. However, as it is something related to the dynamic of the social relations, it is natural that other dimensions should appear as an option of analysis, as it is discussed in Aun [15], the importance of reciprocity as another component of the web.

# 7. The network design

## 7.1. The Agroecological Association of Teresópolis

The Agroecological Association of Teresópolis (AAT) is the result of many decades of learning about the production and commercialization of organic food. The AAT became an association in 2007 but its first step as an organic group was taken in the 1980s. Driven by the ideology of sustainable activity, the idea that organic agriculture could be a profitable business was beginning to emerge among a few enthusiasts

living in the area. At the same time, in the city of Rio de Janeiro, the demand for different food standards began to grow.

During the 1990s, they set up an organizational process to start organic production. Agronomists, veterinary and other liberal professionals started to help transform conventional areas into organic ones by giving technical assistance and attracting new farmers to a new organic production circuit. It was a period encouraged by the emergence of a growing demand for organic food in the cities, in which demand was greater than production. The perception that there was a specific market on the rise was clearly linked to the urban origin of many of the farmers, who at the time still had strong ties to the state capital—home to many of them.

Within this formula, cultivating on the mountains and trading in the city of Rio de Janeiro, the first obstacles began to appear. The first was the growing demand for regular supply. For consumers, it was difficult to understand the lack of standardization and constancy of production. The second major obstacle was the large supermarket chains that, to homogenize the supply of products, ended up imposing prices, packaging and standardization of procedures.

These large trading groups realized the opportunity and niche market organic food represented and imposed on their suppliers the same perverse logic of subordination. This mechanism proved itself unfeasible for the production scale of organic farmers in the region, who, in an attempt to conform to the standards, lost strength and production capacity.

Even after these setbacks, some farmers' urge to support sustainability and organic production in the region did not change, and eventually generated a new form of organization that culminated in 2007 with the creation of the AAT. Teresópolis farmers began to develop another pattern of operation: part of the production would go to the city of Rio de Janeiro and supply street fairs, while the rest would stay in to be sold locally.

At the time of its emergence (2005–2007), the association developed its activities on a very small scale, regarding the amount of food produced and the representativeness in the municipality of Teresópolis and its surroundings. However, it already presented a very heterogeneous character in relation to the types that constituted the first formation of the AAT. It started to grow, concerning the number of associated farmers and the volume produced, after the creation of ABIO's PGS in 2010.

According to the ABIO technicians interviewed, the AAT currently has a group identity and a very interesting configuration, which works well because of the profile of its members. The innovation of this group was the subdivision that occurred within the association. This happened because, owing to the increase of members, the visits to the properties became more difficult because of the amount of time people needed to be away from their crops and the distances. The solution was to create subgroups within the AAT itself, without affecting the association or the relationship between them. For the technicians, this is an indicator of maturity, and for the association, it helped decentralized information and decisions, they had to learn how to communicate better among them.

## 7.2. The participatory guarantee system of Nova Friburgo

In the municipality of Nova Friburgo, the group studied was constituted after ABIO established the PGS system in the region. Before that, there was no organized group of farmers, they were associated with ABIO and their certification was granted after an official inspection.

A social organization movement around a common goal—organic agriculture—was only possible after the creation of this participatory system that, in a way, forces its members to interact in the monthly meetings and visits. This normative procedure eventually brought them closer together and made them rethink their form of organization and communication.

Respondents pointed to a number of factors to justify the difficulty that the group had prior to the PGS to function as a collective, among them, the lack of a common commercialization point—like a street fair—and the distance between production units, which hampers communication and exchange of experiences. After implementing PGS system, they initiated a process of internal change and set up working groups (WGs), which served to aid in other issues besides those related to the PGS process. Respondents were optimistic about the WGs and believed that this process was improving the organization and bringing them closer to a more associative way of working.

For ABIO technicians in Nova Friburgo, the farmers' profile and the group itself were different from the one in Teresópolis. They were more organized around production and less concerned with the associative character of the group. According to them, at the time of the implementation of PGS, there was a history of mistrust among members, which has always contributed to weaken their association. However, it was possible to observe during interviews that the participatory system mitigated this

problem. The innovation perceived was their resilience to build social bonds, even though their seemingly worn out relationship they were able to seek alternatives to (re)organize themselves.

# 8. The perceived dimensions

## 8.1. Social capital

The general idea of social capital is related to the way actors relate among them and how they see themselves as members of the group.

In Nova Friburgo, almost 10 years after the emergence of PGS, the group was beginning to organize itself as a collective. The formation of WGs that sought to organize together the issue of sale, communication and the purchase of inputs were examples of this transformation. Although the distance between farmers has contributed to some isolation, the exchange of experiences was an important characteristic raised during the interviews.

As an obstacle in the process of creating social capital, the excess of bureaucracy and the requirements to be part of the group emerged as important issues, as well as the lack of a common space where everyone could gather and build a stronger bonding. The great possibility of increasing the potential of becoming an association would be the organization of a specific space where they could hold an organic fair. This action demands union and commitment in the process of local articulation, being able to generate new capacities of connections within the rural network.

For AAT members the association's role is clear: to function as a network and serve as a marketing point for its associates' products. In addition, the association makes it easier for them to obtain the organic label, technical support and exchange of experiences.

Although there is almost a consensus about the AAT's role, its deficiencies were felt in the most diverse aspects by the associates interviewed. The process of inspection (for the organic label) and production planning, lack of political representativeness, lack of collective purchase of inputs, lack of communication, the distance between members and the difficulty of reaching a consensus in the meetings were mentioned. Whether by the number of associates (more than 50 at the time of the interviews), or by the group's heterogeneity, the number of questions raised demonstrates, to a certain extent, the degree of involvement in the process of constructing associative thinking.

## 8.2. Endogeneity, sustainability

Endogeneity and sustainability are two concepts that can be put together. The first one represents the capacity of using local resources and transforming them into new possibilities. The second one is how the interviewees relate to their productive areas and the environment around them.

In both groups, it was possible to find many people coming from urban areas, and in some cases, urban areas from abroad. Among the reasons for them to settle in the mountainous region, there were a family inheritance, affinity with the climate and the people, and a place to relax.

Among those who migrated to the region, it was possible to identify, in addition to their interest in organic agriculture, the need to be closer to nature. Nevertheless, the production of food requires both living in nature and having the knowledge of environmental issues. They had an affinity with the region, and with time, they learned how to relate to other rural producers.

Especially in the AAT, it was noticed that the association had an involvement with the municipality by holding their weekly fairs in the urban centre and making space a reference point for healthy food and organic agriculture. This group also hold, as an integral component, local entrepreneurs linked to the gastronomy, tourism and hospitality sector. These two facts contributed positively to the formation of endogeneity between the different actors in the group.

## 8.3. Market governance, institutional arrangements and innovation

These three concepts are related to the actors' capacity to understand the multiple possibilities beyond rural production, their commercialization, their interaction with public policies, their pluriactivity and resilience when facing obstacles.

All farmers interviewed used to sell their production autonomously and all used more than one sales contact. The main marketing strategy was the organic street fairs in their own municipality or in the city of Rio de Janeiro.

The AAT street fair is the most important reference for both groups. It was a space created by the association in the municipality of Teresópolis that operated twice a week. Although it did not represent the biggest volume in sales, it was their most fulfilling sales experience. The reasons for that were the proximity to their productive units; the cultural activities promoted in the venue; the interaction with the public and the lack of standardization or mandatory packaging.

The street fair in Nova Friburgo was mentioned by a few farmers of the city's group, who have managed (with great insistence) a space to commercialize their production. In this traditional street fair, we noticed the presence of conventional and organic producers; other retailers that were not producers; liquor stores and snack bars, among others. It was not an organic street fair, but a traditional street fair that also opened some space for organic farmers.

For the technicians interviewed, farmers always showed a lot of satisfaction with their productive methods and their demands referred to the commercialization. According to the interviewee, farmers wanted more places where they could sell their products without being linked to some formal marketing network and to maintain their autonomy in relation to their products. To that extent, the Carioca Circuit of Organic Street Fairs combined exactly what producers were looking for: autonomy to transfer their products directly to the final consumer.

Regarding the involvement in institutional markets, the AAT began very timidly to participate in public bids for food purchase programmes between 2015 and 2016. At that time, they were discussing whether the productive capacity of members, who agreed to participate in the public call, would enable access to this type of programme. The major obstacle—on the productive side—was the need for joint production planning. The criticism of this model by those involved at the time was that it could limit the farmer within their own work area. In Nova Friburgo, no interviewee reported being part of public food purchase programmes.

Both groups mentioned the difficulty of getting involved with the different public instances of the municipalities. In the AAT, particularly, there was a need for the group to establish a permanent physical space for their street fair and, because of this reality, the political position of the group was stronger and, possibly, more active.

In the AAT, in relation to innovation, it is possible to consider how its members felt the need to reorganize themselves in order to enable a cooperative/associative work among them without disrupting the social structure they have built over the years, in addition to their interest regarding the institutional market, as related above. Here, it is possible to consider the commitment made by the actors involved in the process of managing a collective plan and establishing new flows with the institutional market. It generated in the group a new form of organization and perception towards the governmental programme of food acquisition.

In Nova Friburgo, the moment was of transformation, much more than of innovation. Although the idea of a group was still very much linked to the existence of PGS, some initiatives had already been taken so that the connection among them would be strengthened. Those measures enable the functioning of the group as an association, based, as already mentioned earlier in this text, on the resilience of the actors involved in the process of restructuring the bonds between them.

## 8.4. Final considerations

Two groups were analysed at different moments of the associative process specifically linked to the production of organic food. What got our attention during the interviews and field observations was that, despite the similarities in many physical (climate, vegetation, environment) and social (origin of the actors involved, local economy) aspects, the characteristics of the groups formed, the way they interact with the space, resulted in two very different conformations between them, but that were part of the same network.

To advance in the rural network action, the challenge for organic farmers in the Região Serrana Fluminense lies precisely ingroup functioning, when the individual ceases to be a specific actor and becomes part of an association in which, often, the individual will be replaced by a decision taken by the whole. It is a slow movement of learning and gathering people in a constant exercise of empathy. This process generates a very high-energy expenditure, because there is a constant effort to reach consensus, accept divergent opinions and trust others. In this sense, having groups formed by different types of actors, with different professional activities, but sharing similar ideas, can contribute to a more comprehensive view of the whole.

Within this context of social construction, it is more pertinent to consider the studied universe under the concept of new ruralities other than the individual characteristics of each actor. These new social

conformations in rural areas are responsible for the development of activities of diverse aspects and origins. They have the potential to value (and revalue) local food production by assigning to their products meanings that cannot be imported or globalized. This degree of importance arises from the ability of actors involved in the process of articulating with local capacity to maintain natural resources, generate innovation and strengthen markets and social capital. The synergy only occurs because it is possible to connect people and regions, which proves there are more interactions and interdependencies among the most diverse localities, opposing the dichotomous concept of urban and rural.

Data accessibility. Interviews from field research in Rio de Janeiro State, Brazil submitted to Dryad https://doi.org/10.5061/dryad.w6m905qp0 [16].

Authors' contributions. R.L.A guided the data collection and analysis process in Brazil. N.J.A and R.L.A wrote the text.

Competing interests. We have no competing interests.

Funding. This study was funded by Coordenação de Aperfeiçoamento de Pessoal de Nível Superior.

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
