## [Peer Review File · Royal Society Open Science]

Review History

RSOS-200498.R0 (Original submission)

Review form: Reviewer 1

Is the manuscript scientifically sound in its present form?

Yes

Are the interpretations and conclusions justified by the results?

Yes

Is the language acceptable?

Yes

Do you have any ethical concerns with this paper?

No

Have you any concerns about statistical analyses in this paper?

No

Recommendation?

Major revision is needed (please make suggestions in comments)

Comments to the Author(s)

1. The objective or purpose of the article should be announced in the introduction.
2. The reduced review of literature and theoretical concepts is about pluriactivity (tourism) and new rurality, not about the organic agriculture construction process, and less about the analytical categories or variables of social network among organic farmers and other stakeholders.
3. Methods are not clearly presented, in particular the "multilevel approach and the rural network methodology": what kind of relationships are considered to build the network? snowball sampling used in social network analysis supposed an important number of interviews, but with few questions, about specific relationships. Categories as social capital, innovation, governance or institutional arrangements seems us very wide.

Review form: Reviewer 2

Is the manuscript scientifically sound in its present form?

Yes

Are the interpretations and conclusions justified by the results?

Yes

Is the language acceptable?

Yes

Do you have any ethical concerns with this paper?

No

Have you any concerns about statistical analyses in this paper?

No

Recommendation?

Accept with minor revision (please list in comments)

Comments to the Author(s)

It is recommended to have positions from different groups more clear (see Appendix A).

Decision letter (RSOS-200498.R0)

This year has been very difficult for everyone, and we want to take the opportunity to thank you for your continued support in 2020.

The Royal Society Open Science editorial office will be closed from the evening of Friday 18 December 2020 until Monday 4 January 2021. We will not be responding during this time. If you have received a deadline within this time period, please contact us as soon as possible to allow us

to extend the deadline. If you receive any automated messages during this time asking you to meet a deadline, we offer apologies and invite you to respond after the festive period or during normal working hours.

With our best for a peaceful festive period and New Year, and we look forward to working with you in 2021.

Dear Dr Aun

The Editors assigned to your paper RSOS-200498 "ORGANIC AGRICULTURE AND RURAL NETWORKS IN THE MOUNTAIN ENVIRONMENTS OF REGIÃO SERRANA FLUMINENSE, RIO DE JANEIRO, BRAZIL" have now received comments from reviewers and would like you to revise the paper in accordance with the reviewer comments and any comments from the Editors. Please note this decision does not guarantee eventual acceptance.

Please submit your revised manuscript and required files (see below) no later than 21 days from today's (ie 23-Dec-2020) date. Note: the ScholarOne system will 'lock' if submission of the revision is attempted 21 or more days after the deadline. If you do not think you will be able to meet this deadline please contact the editorial office immediately.

on behalf of Dr Agnieszka Latawiec (Associate Editor) and Agnieszka Latawiec (Subject Editor)
openscience@royalsociety.org

Associate Editor Comments to Author (Dr Agnieszka Latawiec):

Associate Editor: 1

Comments to the Author:

Please incorporate the suggestions of both reviewers

Reviewer comments to Author:

Reviewer: 1

Comments to the Author(s)

1. The objective or purpose of the article should be announced in the introduction.
2. The reduced review of literature and theoretical concepts is about pluriactivity (tourism) and new rurality, not about the organic agriculture construction process, and less about the analytical categories or variables of social network among organic farmers and other stakeholders.
3. Methods are not clearly presented, in particular the "multilevel approach and the rural network methodology": what kind of relationships are considered to build the network ? snowball sampling used in social network analysis supposed an important number of interviews, but with few questions, about specific relationships. Categories as social capital, innovation, governance or institutional arrangements seems us very wide.

Reviewer: 2

Comments to the Author(s)

It is recommended to have positions from different groups more clear. (See attached file for annotated comments).

===PREPARING YOUR MANUSCRIPT===

===PREPARING YOUR REVISION IN SCHOLARONE===

Author's Response to Decision Letter for (RSOS-200498.R0)

See Appendix B.

Decision letter (RSOS-200498.R1)

Dear Dr Aun,

It is a pleasure to accept your manuscript entitled "ORGANIC AGRICULTURE AND RURAL NETWORKS IN THE MOUNTAIN ENVIRONMENTS OF REGIÃO SERRANA FLUMINENSE, RIO DE JANEIRO, BRAZIL" in its current form for publication in Royal Society Open Science.

The comments of the reviewer(s) who reviewed your manuscript are included at the foot of this letter.

on behalf of Dr Agnieszka Latawiec (Subject Editor)
openscience@royalsociety.org

Appendix A**ROYAL SOCIETY
OPEN SCIENCE****ORGANIC AGRICULTURE AND RURAL NETWORKS IN THE
MOUNTAIN ENVIRONMENTS OF REGIÃO SERRANA
FLUMINENSE, RIO DE JANEIRO, BRAZIL**

Journal:	Royal Society Open Science
Manuscript ID	RSOS-200498
Article Type:	Research
Date Submitted by the Author:	08-Jul-2020
Complete List of Authors:	Aun, Nádia; Universidade Federal Rural do Rio de Janeiro, PPGCTIA Assis, Renato; EMBRAPA Centro Nacional de Pesquisa de Agrobiologia, PPGCTIA
Subject:	environmental science < BIOLOGY
Keywords:	organic agriculture, new ruralities, rural networks
Subject Category:	Earth and Environmental Science

**Author-supplied statements**

Relevant information will appear here if provided.

***Ethics***

*Does your article include research that required ethical approval or permits?:*

This article does not present research with ethical considerations

*Statement (if applicable):*

CUST_IF_YES_ETHICS :No data available.

***Data***

*It is a condition of publication that data, code and materials supporting your paper are made publicly*
*available. Does your paper present new data?:*

Yes

*Statement (if applicable):*

The data will be provide as supplementary material.

***Conflict of interest***

I/We declare we have no competing interests

*Statement (if applicable):*

CUST_STATE_CONFLICT :No data available.

***Authors' contributions***

This paper has multiple authors and our individual contributions were as below

*Statement (if applicable):*

Author 1: Designed the studies, collected and analyzed data.

Author 2: Guided the data collection and analysis process in Brazil.

Author 1 and 2: They wrote the text.

**This paper is addressed to the Special Collection on Sustainable Land Use**

Author – 1

Graduate in Applied Social Sciences (Social Communication), with a master's degree in Agroecology
and Rural Development and a PhD in Science, Technology and Agriculture Innovation with main focus
in Public Policies for organic agriculture in mountain environments. My scope of work involves
interdisciplinary groups such as agronomists, social scientists, biologists, economists,
mathematicians, among other researchers and professionals whose main objective is to work for the
improvement of human capabilities in the territory in which they occupy. As a researcher, my main
focus is on people and the relationships established with the environment they occupy in rural or
agricultural areas. I seek to understand how to improve the continuous action of their activities,
opting for models appropriate to the environment they occupy, and building social bonds and
cooperation networks.

Author - 2

Holds a degree in Agronomy Engineering from the Federal Rural University of Rio de Janeiro (1984), a
master's degree in Agronomy (Soil Science) from the Rural Federal University of Rio de Janeiro (1993)
and a PhD in Applied Economics from the State University of Campinas (2002). Is currently a
researcher at Brazilian Agricultural Research Corporation (Embrapa), together with the National
Agrobiology Research Center, working at the Research and Training Center for Farmers of Região
Serrana Fluminense. Professor of the Master's program in Organic Agriculture - an association
between the University Federal Rural of Rio de Janeiro and Embrapa Agrobiology, as well as the bi-
national doctoral program in Science, Technology and Agriculture Innovation - a partnership between
the Rural Federal University of Rio de Janeiro and the National University of Rio Cuarto in Argentina.
Has experience in Agroecology, with emphasis on Rural Development, working mainly on the
following subjects: mountain agriculture, family agriculture, organic agriculture, participative
processes of knowledge building, agricultural sustainability assessment and public policies.

**ORGANIC AGRICULTURE AND RURAL NETWORKS IN THE MOUNTAIN**
**ENVIRONMENTS OF *REGIÃO SERRANA FLUMINENSE*, RIO DE JANEIRO,**
**BRAZIL**

Nádía Jarouche Aun¹, Renato Linhares de Assis²

¹PHD, PosGraduate Program in Science, Technology and Innovation in Agriculture from
Universidade Federal Rural do Rio de Janeiro (UFRRJ), Seropédica (RJ), Brazil (Corresponding
author). nadiarpe@gmail.com

² Researcher at Embrapa Agrobiology, Professor and Tutor at UFRRJ PhD program Science,
Tecnology an Agricultural Innovation, Seropédica (RJ), Brazil. renato.assis@embrapa.br

**Abstract**

This paper presents a case study on organic agriculture at *Região Serrana Fluminense*, Rio de
Janeiro State, Brazil. The purpose was to highlight aspects of local groups involved with this
particular practice. We sought to understand if the aggregating element bringing those groups
together was organic farming while looking into the relevance of other dimensions for this
process. The methodology used to carry out this investigation was based on the concept of the
Rural Network, which enabled us to comprehend the articulations and connections that
comprise the organic agriculture circuit in the mountains of this specific region.

Keywords: Organic agriculture, new ruralities, rural networks

Introduction

Região Serrana Fluminense comprises 16 municipalities, all of them present similar climate and geography characteristics. Their main economic activity is agriculture, except Nova Friburgo, Petrópolis and Teresópolis, which also stand out in the services and industry sectors. The development of organic farming in this region came about during the 1980s and it was consolidated at the end of the following decade, thanks to an alliance between agronomists and traditional local farmers who saw in the local mountain environment an ideal place to build a new way of life associated with a new way of doing agriculture.

Currently there is an average number of organic farmers organized in associations, groups or companies in the region with the purpose of producing and marketing food. Even though these associations are outnumbered by other rural producers of the region, they can also count on the knowledge of different types of professionals: from lawyers to chefs, those organic groups have developed a special way to establish a connection across the various regions where organic food is present.

We have focused the present article on organic rural workers from Nova Friburgo and Teresópolis. Firstly, we contextualize the research. Then we present the methodological procedures used to identify the rural network. Finally, we depict actors, dimensions and how they are organized around organic agriculture and how they are contributing to institute a more complex rural network.

Teresópolis and Nova Friburgo

Rural production and tourism are common economic activities between Nova Friburgo and Teresópolis. Rural production has always been a local aptitude, although the territory faces restrictions to mechanization and difficulties to install a production flow. The mild climate (tropical of altitude) with rich vegetation cover (Atlantic Forest) and year-round high rainfall has allowed the region to establish itself as the green belt of Rio de Janeiro, the capital and the metropolitan region.

The properties are small in size and the predominant crop is greenery (ASSIS et al, 1996). This region's agricultural production accounts for 90% of the vegetable supply (CARNEIRO e ROCHA, 2009) for the 12,377,505 inhabitants (IBGE, 2017) of the metropolitan region of the city of Rio de Janeiro.

Pluriactivity was identified as a unique feature regarding the actors from the region. Marafon and Ribeiro (2006) and Carneiro and Rocha (2009) have observed the existence of farmers who had other roles, both in urban centers and within their properties, which were chiefly related to tourism. There is, however, the perception that the importance of agricultural activities in the composition of household income is decreasing, nonetheless, it does not mean it will disappear, since its paramount importance to maintain the identity of the population and the territory through the sustainable use of natural resources (CARNEIRO and ROCHA 2009).

[revised manuscript text omitted]
 <http://www.agricultura.gov.br/desenvolvimento-sustentavel/organicos>. Acesso em: 08/08/2018
- MARAFON, G. J.; RIBEIRO, M. A. 2006. Agricultura familiar, pluriatividade e turismo rural: reflexões a partir do território fluminense. **Revista Rio de Janeiro**, n. 18-19.
- MATTOS, R. A. 2010. População neorural e agricultura orgânica: mudanças no meio rural da região perimetropolitana do Rio de Janeiro. **Anais.. XVI Encontro Nacional de Geógrafos**, Porto Alegre.
- PERUZZO, C. M. K. 2017. Pressupostos epistemológicos e metodológicos da pesquisa participativa: da observação participante à pesquisa-ação. **Época III**, Colima.
- PLOEG, van der J.D.; MARSDEN, T. (eds.) 2008. **Unfolding Webs: The Dynamics of Regional Rural Development**. VanGorcum,
- SOARES, M. D. O. 2007. As Contradições do turismo no espaço rural: vida, trabalho, renda e exclusão. **Tese de Doutorado**. Feagri – Unicamp. Campinas.
- VINUTO, J. A. 2014. Amostragem em bola de neve na pesquisa qualitativa: um debate em aberto. *Temáticas*, Campinas, 22 (44).

Interview Jorge (the Swiss) - 02/29

N: So come on Jorge, first I would like you to start by talking a little about your history and your relationship with Teresópolis

J: I was born in 49 in Switzerland, a very beautiful city, it was a city, but I always had contact with the countryside, our family has a house in the countryside, because my father came from the countryside too, he was not a farmer, he was a doctor, so I made a very different profession than I am doing now. I was a hotelier there in Switzerland. We had a project all over the world. With this work I traveled a lot. A different trip ... luxury hotel, 3 star restaurant ... but always with food.

So at the age of 40 I decided to do something else now, I sold everything and bought a backpack and started traveling. I crossed the Atlantic with a sailboat. I arrived in the Antilles, I went down to Brazil, I traveled through South America all back and when I went back to Europe, my father passed away. I helped my mother fix things and then continued on to the east. And only Africa was missing but then I was tired and I went back to Switzerland. There I went to visit a friend who had a property in the mountains in the south of Switzerland, I asked you do you need someone to work and he said he didn't need it but that he had a job. He had no money to pay but he offered a house, a place to sleep, food and such. And when you need ... ah tomorrow, so I went. An area of 6 hectares, very well organized. Ah, during the trip I took, I always worked with agriculture, harvested cocoa in the Amazon, planted rice in India, took care of cows in Australia. It wasn't work for money, it was for food and a place to sleep. Well, I got there on this very well organized farm, a small family, the children had already left, just the couple, he had 30 goats, an orchard, with lots of apples, pears, cherries, kiwis, persimmons, it was in the south of Switzerland with a well-tempered climate. A vegetable garden with various types of vegetables: lettuce, carrots, parsnips. Everything organic and very well organized. We had grapes, it was a wine growing region. Very mountainous and all his agriculture was done on a terrace, a terrace made of stone, very old. Every region. In the summer we went up with the goats on the mountain, two hours up, they stayed up there grazing. We express milk in the morning and then in the afternoon and let go ... and they went up, up. This for three months, because the summer is very short.

Well then I came to stay for three months and ended up staying seven years. Then I ended up developing a disease due to milk, which already existed but which started to become very active. Then I decided to leave. As I already had a sister who lived here for 40 years she always invited me to come. She already had a place, I even knew it. So I made that decision 20 years ago, that was in 1997. And here I met my future wife and then it made sense to buy us a place. So I bought this place here despite the idea of self-sustainability being very big ...

With that friend of mine in Switzerland, we didn't do a fair, customers came to buy at home. Then I too the idea was not commercial, it was sustainability. This place was once all productive. A former Dutch owner, had a large plantation with his own truck, took everything to the city, then the children grew up, all moved to the city, the owner died and the children decided to sell. A real estate company bought and left it for 30 years. So when I arrived I wanted to buy bush, and I helped plant more trees and also planted my food, much smaller than now and I had a cow. Then came the idea of making the organic fair in the city.

Then I found Beto, because Beto was the oldest in the group, he came from a young university,
I came to buy this place where he is now, very young, already with a daughter and they lived
there in the country. He sold among the conventional ones at a fair near Bus station he sold his
products, then after about four months I saw a course on medicinal plants I decided to do to
learn more about local plants, in this course I met Ana Litardo who also does juices today at
the fair. At that time little spoke Portuguese either. I also met my wife on this course. I
negotiated five years to buy this land here, it was very expensive, I managed to lower it to half
the price. So as soon as we arrived and bought and settled everything, I started to open a bed
to plant, because that was my idea to have my own food.

In the not very good soil in the productive sense, because it is deep in the valley, very sandy,
little deposit of good land, the water took everything. You know how water works, you know,
water always goes where it's easy to walk. In the past, this river had much more water, today
silk and it takes 70% to the city. Straight from the source. The best water is in town, because it
is born in the middle of the forest. So I started to plant and compost. I compose a lot, I like to
compost. That's why there were cows here. I don't eat meat or drink milk, what interested me
was the manure! And then I learned how to make the compost dynamized, so I stopped
needing that compost.

27 N: how was this meeting with Beto?

31 J: and soon we saw that we had affinities of thought. He worked for the market right? There
were sharecroppers and such, they planted commercially and everything, already organic.
There were always problems with these sharecroppers, people who came from the
conventional, so we started talking more about planting just for you and such ... and now he
has this guy who is very good, he counted a lot with Gustavo until two years ago. This boy who
is with Beto Fabiano is also a tractor driver, he knows how to handle the machines very well.
Although he came from the conventional, he also came to Beto for health reasons. Well, we
formed this association in 2007 and started with a fair near the city hall, a private area for
some of Beto's friends. And then that area was sold and we had to leave. And he arranged this
other place that we are there. So this area is not ours ... for us it can create a problem ...
because the owner of the area already has a project and wants to urbanize the place ... so
soon we need to get out of there.

48 N: is the relationship between aat and the city hall turbulent or harmonious?

52 J: you know our city hall I don't even know if it exists ... teresópolis history is very confusing.
The current mayor who has been mayor in other times is not a good person and he is perhaps
the fifth mayor of this mandate ... and it doesn't work to find out if he will continue to be
tomorrow ... and this is a terrible delay for the city, because nobody invests in the house,
nobody invests in transit, in the dump that is already full. The municipality is bankrupt ... they
took the 3 million employees' pension money ... stolen !!

1
2
3 N: the city could even use the agroecology fair to publicize it and such, doesn't it?

7 J: No ... we even try to relate to the agricultural secretaries but every 2 or 3 months it changes
and there is no money or personnel to help. The tourism secretary is the same.

11 N: today looking back on the history of the association as you built it, do you have any role in
the association / association that is important for you in your opinion?

16 J: perhaps the most important thing is the creation of a society. It is the creation of a society
the work that each one is responsible for is very fundamental that is lacking in our society so
an example with PGS, which incidentally raised the association forward, each one has
responsibility for the functioning of this system, because we shared it when aat grew a lot,
divided into 5 sub groups and each sub can have a maximum of 12 participants, because 12
because the year is 12 months old and each sub group makes a verification visit at each
property of its sub group once a month, once exchange of experience between producers that
has never happened before in this intensity. So the quality of the product increased a lot, so
did the knowledge of the producer. Why are you going to visit the neighbour you met but
never had a technical chat at that intensity. You will visit the neighbour and talk to him about
the carrot that is better or worse and exchange experiences and techniques. This for me is a
phenomenal discovery, Brazil is the only country in the world that makes this PGS system, I
don't know anywhere else, that I know of. Because the audit, it is also a guarantee but it does
not have this exchange and nor can it have, it is not allowed. The auditor goes there and
discusses with the producer and asks and such and say yes or no. And our business is different,
agent causes discussion, we have to discuss. So just one example, last week, my group came
here and another group was scheduled to go to another property, but it was nobody, just a
producer. And the producer was really upset and such and called me saying what had
happened. Then I went to check what had happened and the producer herself felt that the
responsibility for calling and organizing the visit rests with the representative of the group,
when in fact the responsibility lies with the person who receives the visit. So she would have
had to get in touch with everyone and organize everything. So I spoke to her and explained
how it works and we scheduled another visit. And they all went. Only when everyone takes
their responsibility will it work.

48 N: do you think there is any other possibility for aat to act beyond this contribution in the
49 formation of more socially responsible individuals?

53 J: I think that social formation is the most important thing

57 N: and when it comes to trade, do you think that the association has a role in marketing each
58 farmer's products

1
2
3 J: yes, there is, because until now we have practiced the free market, we do not give
instructions or rules about prices and each one makes his own price. And that generates
criticism. For example, cassava, one sells for 5 reais a kilo, the other sells for 4 reais and I sell
for 3 kilo. What's up? What happens, I think 3 reais is reasonable, because the market price is
between 1.30 and 1.90. so we don't want an expensive organic product, we want an affordable
product, because we have to win the average consumer. But funny that the poorest of these
farmers tores sells the most expensive. We will discuss how you want this. I want a fax to limit
prices and others don't ... so this is a subject that is ...

15 N: and do you notice from the beginning of the fair until now if there is a change in the public
at the fair? you managed to reach this common audience?

20 J: the audience has certainly changed. we have at least 10 times more audience than at the
21 beginning. Producer has increased, we have 30 stalls at the fair and in five years it is expected
to come up to 100 stalls and invade part of the parking lot.

26 N: can you point out any weaknesses of the aat?

30 J: ah always has. Participation in the organization's work still needs to be more participatory.
There is a clear training problem that will influence ... I am, for example, a manager at the fair
and aat, so to do that, you have to have a little knowledge and the ability to handle documents
and such. I am looking for someone like that now, as I have been doing this for some time now
and would like to get some rest.

39 N: and do you consider yourself a paid person to do this job?

43 J: No. It has to be a partner. You can even share, but you have to be a partner. Else who will
pay? Our coordination is shared by 5 hugo, Ana, Beto (cultural), Jorge (Adm and financ),
Valentine (marketing), João Gallo (Technical Coord.). This helps with coordination and we have
a vote in April which is a general meeting to change positions.

50 N: you work on some aspect of planning your production.

54 J: this is an important issue. And to make it really work, the agent association needed to do
some planning. Agent still hasn't managed to make a plan generates and as several
associations participate in fairs in the Rio circuit and nobody plans for everything they need to
sell so the exchange of product is important and always have the right prod at the station but
until now we haven't been able to install planning. I have a plan following the seasons, but it
was disturbed by the invasion of jacu ... So I'm kind of poor in products now. What I am going

to take to the fair now is little: it is cassava, sweet potato, taioba, banana, carrot, silver, but
this is usually not much, I have a lot more product, peas, no cabbage, cabbage or cauliflower. .

8 N: do you have any other selling points?

11 J: here at home. I sell more honey here than at the fair. I produce between 500 and 600 kilos a
12 year. Between 15 and 17 hives.

16 N: do you work here with any help?

19 J: yes I have an employee who has been with me for 19 years and a daily cleaner for 6. this
oldest employee was a construction assistant in the construction of my house. He had a farm
but he sold it to build his house here on my land. And he's been with me ever since. A very
good, very serious employee.

27 N: do you think that this commercialization process within aasoc can change or take a leap and
28 you have another sales channel, like PNae or small attackers?

32 J: yes, there are many markets. You just need to produce with quality and at a good price.
There is no point in coming with crazy prices.

38 N: have you used any technical assistance?

41 J: agent has 5 agronomists in the group. One is Beto, Ana Paula who became responsible for
the SPG at abio. She is a collaborator of the association. She brought spg to us. He worked for
abio and they came to talk to us, because neither Beto nor I had much interest in certification.
There in Switzerland I didn't have a certificate. I know it's organic, you don't need a certificate.
But then the legislation changed and if you want to sell organic product you have to certify it,
so let's do it, let's do it ... then we did three or four meetings and decided.

Farmers who live exclusively on production:

Avenir (also indicated by Beto)

Renato (it's another dimension, full of partners, very organized, many customers)

João Gallo lives on that too. Our seedling and compost and agrobio producer.

Marquinho (works near Avenir is a street marketer like Avenir, but he only has fairs in the river
and he and his wife do six fairs a week. He has his own truck and five employees who work
with him, 8 greenhouses with only strawberries)

Gustavo Interview - 03/24

G: This movement from AAT must have already told you that it came from the fair. First the
fair came up with a group of producers, it was a very weak movement at the time, the organic
was forgotten for many years since there was no fair, no marketing network in the region and
we started the fair in a nearby store and it was a little place of producers, some had already
participated in the movement for a long time like Beto who participated in abio, and already
participated in other fairs in the region. And other smaller producers who were starting with
the organic thing and it was a movement that grew and there was a moment when it was
necessary to form this association. I see that this is the same movement that continues to
expand and deepen more and more and we are in a very beautiful moment very strong very
deep in several issues still has a lot to walk and structure but it is working well in the middle of
this very urbanized city and on the other hand it has a very strong and intense agricultural
activity in the conventional sense and is gaining strength despite of the difficulties, political
difficulties, the space itself the question of the space of the fair that does not belong to us.

21 N: What is your history with the region? Why agriculture?

G: it was the path of my parents who were from rj and my father graduated in biology and
wanted to leave the city and chose to live in a sinkhole, soledad a municipality next to
Teresópolis, a completely agricultural municipality with no urban center, an area totally turned
to agriculture. And I grew up there until I was 6, 7 years old when I moved to the city with my
mother and sister to study. But my father always had the place and he never stopped and the
call came from a child, I grew up in the middle of the country, eating things and my taste was
cultivated from the beginning and as soon as I finished high school I went straight back to the
countryside and I tried to work with my father one year but I felt that it was not the way I
wanted in the way of working, more entrepreneurial and production focused on Rio de
Janeiro. I was looking for something more ideological in terms of having more food in my own
home, more self-sustainability, a different form of marketing too and that was when I met
Beto and he was in a moment of transition in his life, he had separated and was looking for a
new way of production and agent joined and we lived together for 10 years. And it was a very
rich moment of research, very laboratory to try things, to dive a little into ideology, a little
freedom to be able to experience a way of living and a way of producing that was rich and
reflected a lot in the ideology of the fair and the association.

43 N: this is what I was going to ask if you think it influenced ...

G: it sure is very rich. Today, the group at the fair is extremely heterogeneous, but there is an
agroecology line in search of something more essential for direct sale of more genuine
production. I think this is present and we lived there for 10 years and started the fair together,
beto and others. producers, Jorge was not yet because he did not have significant production.
He comes in after and behind this administrative structure, which is very important for the
functioning of the association and has grown.

53 N: and do you see this thing of aat being more and more structured and formed as something
negative in the sense of maintaining this thing of ideology and care for the environment and
people, with the production that would be a bit of the tripod of agroecology?

G: not quite the contrary, I think it supports and supports the ideology when you have a group
that knows how to discuss the issues that knows how to make collective decisions and have
the strength to implant your ideas with the most foundation and strength for ideology.

1
2
3 N: do you work with production?

G: no, after a while i left beto's place. And I work with something, I have a connection, I'm a
beekeeper, I do other activities with the environment, I prune trees and I'm a guide in the
national park. But I feel like a farmer and I feel that it is my way but I am in a moment of I
don't have a land to live on, I would like to have the means to acquire a reasonably sized land
to live and plant my food. It is what I intend, I feel like a farmer and that my job is to be on the
land.

13 N: the question of land tenure is a difficulty for young people today. Do you think this is an
14 issue? Do you know alternatives? Do you see this as a failure in our process of supporting the
15 young producer?

G: i totally think if i had the means to acquire a land access to viable financing i would be
producing there. This is a fact. I lived for many years in a land that was not mine and it was
very good, important, I learned and built a lot but you dedicate your life, you build things that
you need guarantees that you have security over that it is no use you just live in someone's
land and you may want to be just a farmer but build a life a family, work with something other
than horticulture, but as an agroflor this things with longer cycle you have to have a guarantee
of the land, this is extremely important I know that there is a funding for pronaf, but very
difficult to get information and have access to it. I tried to search but I didn't get it. It seems
very bureaucratic and complex the owner of the land has to be very in order to sell to you in
this line because it is slow. I think it is a fundamental question. You will not buy land with
agricultural work, it is very expensive, and the reality is that you do not pay for the land with
her work. you pay her being a businessman, having an inheritance, and working in the city.

32 N: how do you see aat today? With that number of producers?

G: first I am very happy to see that there is something really cool going on and an expression of
real work going on in many branches, many different activities, an important force for our
region at different levels, a fundamental political force. I hope that we will be able to maintain
this pace and continue to build and consolidate what is still necessary because it is very
unstable and insecure, but it is a way for us to have political strength and achieve some
transformation in our society, in our way of life I think it is a mechanism a means for that. The
end result is the association itself already a tool for transforming reality. Certainly, there are
people here who are not satisfied with how urban development and social organization are
going.

46 N: and in what sense is it a transformation?

G: complete way of living, way of relating, way of marketing that involves several different
situations between dealing with the public and the product

51 N: what about the question of price in marketing? Do you think there needs to be an education
of the public or a reduction in prices?

G: it's been like this for some time that I haven't produced and I feel distant from the
consumer of the fair. Although I always come to play every Saturday, but this part of the
commercialization has been a long time since I've been away. But while I was there, the
presence of simpler class people who were housewives and maids who attended the fair and
valued it was very strong, showing that in fact we were providing viable products not only for a
privileged class but to acquire. But for sure there is still something there ... but as people get in

touch with direct sales in our more specific case, I think it is still viable. In the larger market it is
already possible ... the price has a greater impact. And an important thing is education like
understanding why some products have a higher price and there is no way not to be. I think
this still lacks a little information.

8 N: are you a consumer of the products?

G: not buying. As we have the project at the fair, one of the contributions is a basket of
products for each musician. It is a very generous basket. It's an exchange

13 N: how do you see the role of aat for producers? Especially after the installation of the PGS

G: I think she is fulfilling her initial goal. That since the beginning, the statute that people
formed that was to strengthen the agroecology to make this movement grow is fulfilling it. The
result is that you arrive here at the fair and see the amount of fairs and products that you have
here and that is the direct result of the work you provided. Having a sales space here at the fair
with a certain firmness is a crucial factor for a person to become organic. If you are already a
producer and decide to enter the main conversion question is how to maintain your economy.
Everyone already has a scheme to send products to Rio, ceasa the truck that goes down and
everything and how does this transition. So this consolidated selling point. Where a person can
go and continue to develop their economy. Not just the fair, actually. Aat involves various
forms of marketing, you know, there are people who go down to the river and make baskets
and do fairs on the river. Here, it is like a center that has the fair, but it also has other forms of
commercialization that happens through the association. In addition to the support that aat
offers, lay people arrive and quickly start to learn.

32 N: did you participate in the process of switching to PGS?

G: yes, it was. It was very rich. Now a days PGS is a very special thing. The interaction that
creates the exchange of knowledge between social producers, even though sometimes the
person spends his whole life in production every day, comes to sell and does not create the
bond, does not visit the neighbour. PGS provides this. Before I didn't have that exchange, I
came to the fair and one or another meeting. But now they meet all month, exchange
technical information about production and such

41 N: and do you see any opportunity from aat that she is not currently doing? that it can become
an evolution a new path ...

G: certainly there are many, there is much to grow. I think on the political issue, on
representativeness within the city. There has to be more political strength and strength to
transform more. Production has great potential in this region. And most of our products are
sold on the river and there are a lot of consumers here. In the area of science and research,

50 N: can you see any flaws in this process? How do people deal with aat?

G: it is all group organization, certainly one of the main issues is the involvement of people and
ensuring that everyone takes on their duties. Every social org I think has that. But I wouldn't
know how to point out a big flaw. There are many small flaws that you learn and repair and
build.

57 N: would you be able to say how the people of the city see aat?

G: I couldn't say, but I believe that we have a good image. Because we have been participating
in various political activities and city councils for many years and with the participation we
have been gaining respect. The beto himself was representative of councils here that I can't
remember the name. But finally it showed that there is respect and serious work ...

9 N: would you know to give me some advice to talk?

G: I couldn't say. I've been out of it for a long time. They have been very deflated. And I can't
say which ones are currently active. Teresopolis has been experiencing political turmoil in
recent years. We have a totally unstructured policy. And I don't know how the councils are
working today. Now we are with a mayor who had been elected, but he was ineligible, due to
political manoeuvres but then the last mayor was tired and he comes back on the scene ...
anyway ... and then it impacts directly on the activity. For you to see, in the first month that
this happened they came and took all of our plants here. For you to see, we are totally
vulnerable here in this field. Last year there was news that we needed to leave, we had to get
organized fast and such. Then the luck that some lawyers appeared that helped us in this, we
won a title of public utility for the municipality, with the articulation from this episode. But it is
actually still unstable and insecure. It is a private land of this factory and that the city uses the
land in return for some debts from the factory and the city has assigned a part for us to use ...
but that has nothing written, no paper at all. we believe in growing and gaining a lot of
strength and never leaving here.

28 N: have you tried to talk to the factory owner?

G: yes, we contacted the land attorney and learned that they knew nothing about this pressure
and the need to leave and then the business died. Nobody talks about it anymore.

Interview Otávio Miata

Otávio Miata, my history as a farmer begins in Santa Cruz, my family (part of the father) is all about the colonization there, then in 2014 this uncle of mine (his son who owns the farm) he already developed organic farming there in 2009 site. Then he had a problem with his spine, so he didn't want to give up all the investment and called me to continue the work. Then I accepted and in 2014 I came here. In Santa Cruz he worked with manioc. I have been in this struggle since 2014, trying to survive on organic agriculture. In fact, I would put deadlines ... if in three months there is no return I will give up, then when the 3 months came I would speak another 6 months ... then when the 6 months came I was still in the red and said: I need to wait for another year, and then I stayed. It still doesn't come out of the red very well but at least I see some perspective. But it takes time to give a return and the investment is heavy (there is a lot of investment, a lack of resources and development is slow ...). And the main factor is that I really like this city (the climate, the people) and I have no intention of going back to Rio. And it is an activity that does not give me much return ...

Luã Madureira (advises the city government on environmental management is taking this opportunity to implement agroforestry techniques in the management of neighbourhoods) - I think she deserves it too, we have so much conventional production that I think it is a city worth the investment to try change that outlook. The roads pass inside the land I'm Luã Madureira, and I still don't consider myself a very farmer because I'm at the beginning of the project but a friend of mine who already has experience with agriculture was looking for land wanting to start this project agroforestry, and soil improvement and he came to me knowing that I had a piece of land, a place that was more of a leisure that we are now renovating to have a production really. There are still three years of investment, there is little time for planting itself, and as Otávio said, it requires investment of everything that I do not have this capital as easy as that, nor time, because it is necessary to work abroad to be able to bring this capital in. of the site. I can't drop everything and stay in place. There is a friend of mine who says that I am a modern hermit. That I want to drop everything and stay in place, but unfortunately it is utopia to drop everything and stay in place. Because we need to survive. But the objective is to spread this a little, Friburgo deserves it ... the city has a lot of conventional farmers who do not believe that organic agriculture can reach a scale of production that meets the demands of the market. They say thus: iii without poison it is not possible. This is one of the factors that motivates me to make this investment ... prove that it is possible ... When we started doing our planting together with eucalyptus and bananas, the nursery technician said that although we are organic and such in the eucalyptus needed of poison because it wouldn't work ...

Otávio: when your organic farm develops, with variety with biodiversity, you can see that the ecology is really working, your system stabilizes itself in such an impressive way. So the importance of investment time ...

Nádia: how is this story of the group, of the association?

Otávio: Ah ... there it is with Marc or Luis Felipe

Nádia: But what is your perception?

Otávio: RJ itself does not have a cooperative or associative culture. So much so that assoc or coop here ... not to mention ..., but you go to SP or RS, you leave RJ and already see many institutions like this working. My perception is that RJ does not have this profile ... But speaking

here, it had a boom, a growth and got out of control and we are trying to get around this, with
some processes, everything comes up against the value to keep everything working well,
including investment in the process. Everything takes money. But ABIO has a strong vision of
association, and not only as certification. Our group is Nova Friburgo and region, which
includes some municipalities in the mountainous region in addition to NF and is within ABIO,
which is an association of more than 500 or 600 members who are divided into groups. We are
17 certified members and some collaborators and others who are looking to join, as is the case
with Luã.

Nádía: What is the group? Do you see yourself within the group as something beyond
certification?

Otávio: yes, including our group, when Marc was leaving, he was the facilitator, he had the
idea of creating the working groups, so only in that you can see that he is beyond certification.
And our debate in the field and in the meetings themselves you see is very rich. There's a lot to
talk about besides certification. It is not working very well, but we have five working groups
here in Friburgo: participatory generation of guarantees, collective purchases, marketing,
communication, and the technical issue that is the productive approach. Each working group
has a head. In the case the head of the productive approach is having to make a seminar, open
to the public, telling the story of ABIO, and talking about the main concepts related to our
practice: ecology, agroecology, organic agriculture. So I think that beyond certification we are
doing our job.

Luã: I'm more out of the picture and now I see that the bottleneck is marketing, there's no
fixed point for marketing. It has nowhere to be marketed and so one of the points within the
marketing group is dealing with this longstanding problem for all producers. The cool thing is
that we are trying to solve this together, instead of each one solving their problem alone.

Is the commercial issue more individual or collective?

Luã: now that we are really trying to bring the focus of the issue to the group, collectively
resolve the issue of commercialization. But it is still within the discussion groups ... working. In
general, everyone sells / markets their products individually. I really think what is missing is a
better check of conventional plantations. I particularly consult with the city in an area that is
re-technical but despite the lack of knowledge (I am not an agronomist or technician) I have
the experience of farming and living in the countryside. I am then organizing some lectures and
seminars on the topic and trying to develop sustainable vegetable gardens in the
neighbourhoods.

The public thing is something that occurs much more at the local / regional level and then it
radiates ...

Luã: yes, yes without a doubt. From here to there and from there to there. Who has this
facility that is articulating locally and making their contacts ... but it is really very difficult. We
see it is not so interested. By public agencies we see that the conventional is increasingly
stronger and if you want to with poison you will plant it ... but we are trying to swim against
the current spreading this activity.

Otávio: the collective purchase is for the purchase of inputs ... We buy everything here.
Seedlings are more difficult ... it's all conventional. Most of the group's farmers try to make

their own seedlings, but when they can't, they end up buying from conventional ones. Organic
seedlings are very difficult here. I can't tell if anyone in the group accesses PAA or PNAE.

Do you believe that some kind of policy would be interesting to encourage the beginning of
organic activity or even its continuity?

Otavio: oh this is fundamental ...

Luã: the beginning is the most difficult ... if you don't have the dedication and the focus the
business will not go forward. Because there are many barriers. Even with the terrain, there are
some barriers beyond the difficulties with the initial investment, here in Friburgo for example
not everyone wants to help themselves. Here there is still a lot of thought that when one wins
the other loses ... it is difficult for people to understand this win-win relationship.

Otavio: this is what I mean with a lack of associative profile.

Luã: Within our group, we try to make everyone win. In society, there must be an awakening of
conscience ... it is no longer possible to win and the rest to lose.

Otavio: this is the SPG system, the social side of warranty verification.

Luã: yes ... anyone outside the line harms the whole group !! Nobody wants someone outside
the line within the group. This creates trust between participants and consumers. A
relationship of interdependence between everyone begins ... farmers, consumers and the very
abio that ends up becoming stronger.

Nadia: is there something you guys are not doing and you need to do ... a project for the
future?

Luã: What would be the group's second step, otávio?

Otavio: I think this is the business of marketing ... We need to develop a lot. Each has its
channel, its point, its home delivery ... each one is on its own. We also have to organize an
organic fair, even though I think we have difficulty finding space. There is the fair in the village
amelia, which is conventional, I have a bank there, me and Mr. Bicalho have a bank there. That
space is municipal provided for a cooperative and is very disputed !!! It is a very traditional
space, which the entire population already knows. Older marketers don't want to leave. The
group's producers deliver baskets in RJ, and fair in RJ, I particularly, also do the fair of the
village amelia and baskets here.

Luã: I'm from the communication area and I'm in the communication gt. I try to open new
sales channels and take care of the dissemination on social networks.

Otavio: I'm an agronomist at Rural and I bring the technical part as luggage, but I can say that
agronomist when he leaves college and goes to the countryside he gets lost ...

Appendix B

Dear Editors,

In response to Reviewer 1

1. We removed the purpose of the abstract and placed it in to the Introduction.
2. Yes, it is correct. We did not discuss the construction of Organic Agriculture per se it was not our main purpose but how it became a converging point among the different kind of actors that were living in the localities we researched. We discussed, briefly the dimensions of the rural network in the Methodology section. We also prefer to use the term 'actors' instead of 'stakeholders' since we understand 'actors' has a greater amplitude of meaning and is more usual within social sciences. We also believe 'stakeholder' is more restricted to economic contexts.
3. We considered any kind of relation valid as a way of creating bonds between actors, groups or organizations. It can be based on income, knowledge, reciprocity, similarity of any kind. The idea of the rural web is how it is possible to connect people, places and activities of various origins having as central point Organic Agriculture. With that in mind, we started from the groups that were already structured and from them understand what kind of bonds are being built among everyone involved. From the farmer to the chef.

The categories you mentioned, which we prefer to call dimensions, are very wide indeed. Therefore, we had to use more of our participatory observation to understand how we could use each one of them as well as to create indicators to help us define boundaries between those dimensions.

We did not discuss the dimensions (concepts or specific definitions) in the Methodology section but in the Network design section, where we related the dimensions defined in the rural web methodology with data collected in the field.

In response to Reviewer 2

1. We accept your observations with one exception: As explained above, we prefer to use in this paper the term 'actors' instead of 'Stakeholders'.
2. We highlighted in yellow all changes made in the original text.

Kind Regards,

Nádia Jarouche Aun

Renato Linhares de Assis